# Decreased Passive Immunity to Respiratory Viruses through Human Milk during the COVID-19 Pandemic

Marloes Grobben,[a] Hannah G. Juncker,[b,c] Karlijn van der Straten,[a] A. H. Ayesha Lavell,[d] Michiel Schinkel,[e] David T. P. Buis,[d] Maarten F. Wilbrink,[b] Khadija Tejjani,[a] Mathieu A. F. Claireaux,[a] Aafke Aartse,[f] Christianne J. M. de Groot,[g] Dasja Pajkrt,[b] Marije K. Bomers,[d] Jonne J. Sikkens,[d] Marit J. van Gils,[a] Johannes B. van Goudoever,[b] Britt J. van Keulen[b]

[a]Department of Medical Microbiology and Infection Prevention, Amsterdam Infection and Immunity Institute, University of Amsterdam, Amsterdam, the Netherlands

[b]Department of Pediatrics, Amsterdam Reproduction & Development Research Institute, Emma Children's Hospital, University of Amsterdam, Amsterdam, the Netherlands

[c]Center for Neuroscience, Swammerdam Institute for Life Sciences, University of Amsterdam, Amsterdam, the Netherlands

[d]Department of Internal Medicine, Amsterdam Institute for Infection and Immunity, Amsterdam UMC, Vrije Universiteit Amsterdam, Amsterdam, the Netherlands

[e]Center for Experimental and Molecular Medicine, Amsterdam Institute for Infection and Immunity, University of Amsterdam, Amsterdam, the Netherlands

[f]Department of Virology, Biomedical Primate Research Centre, Rijswijk, the Netherlands

[g]Department of Obstetrics and Gynaecology, Amsterdam Reproduction & Development Research Institute, Amsterdam UMC, Vrije Universiteit, Amsterdam, the Netherlands

Marloes Grobben and Hannah G. Juncker contributed equally to this article. Author order was determined alphabetically.

**ABSTRACT** Infants may develop severe viral respiratory tract infections because their immune system is still developing in the first months after birth. Human milk provides passive humoral immunity during the first months of life. During the COVID-19 pandemic, circulation of common respiratory viruses was virtually absent due to the preventative measures resulting in reduced maternal exposure. Therefore, we hypothesized that this might result in lower antibody levels in human milk during the pandemic and, subsequently, decreased protection of infants against viral respiratory tract infections. We assessed antibody levels against respiratory syncytial virus (RSV), Influenza virus, and several seasonal coronaviruses in different periods of the COVID-19 pandemic in serum and human milk using a Luminex assay. IgG levels against RSV, Influenza, HCoV-OC43, HCoV-HKU1, and HCoV-NL63 in human milk were reduced with a factor of 1.7 ($P < 0.001$), 2.2 ($P < 0.01$), 2.6 ($P < 0.05$), 1.4 ($P < 0.01$), and 2.1 ($P < 0.001$), respectively, since the introduction of the COVID-19 restrictions. Furthermore, we observed that human milk of mothers that experienced COVID-19 contained increased levels of IgG and IgA binding to other respiratory viruses. Passive immunity via human milk against common respiratory viruses was reduced during the COVID-19 pandemic, which may have consequences for the protection of breastfed infants against respiratory infections.

**IMPORTANCE** Passive immunity derived from antibodies in human milk is important for protecting young infants against invading viruses. During the COVID-19 pandemic, circulation of common respiratory viruses was virtually absent due to preventative measures. In this study, we observed a decrease in human milk antibody levels against common respiratory viruses several months into the COVID-19 pandemic. This waning of antibody levels might partially explain the previously observed surge of hospitalizations of infants, mostly due to RSV, when preventative hygiene measures were lifted. Knowledge of the association between preventative measures, antibody levels in human milk and subsequent passive immunity in infants might help predict infant hospital admissions and thereby enables anticipation to prevent capacity issues. Additionally, it is important in the consideration for strategies for future lockdowns to best prevent possible consequences for vulnerable infants.

**KEYWORDS** antibodies, RSV, Influenza, coronaviruses, breast milk, COVID-19

Address correspondence to Johannes B. van Goudoever, h.vangoudoever@amsterdamumc.nl.

The authors declare a potential conflict of interest. J.B.vG. is the founder and director of the Dutch National Human Milk Bank and a member of the National Health Council. J.B.vG. has been a member of the National Breastfeeding Council from March 2010 to March 2020.

The coronavirus disease 2019 (COVID-19) pandemic rapidly became a public health emergency of international concern influencing public health, society, and economies worldwide. Since transmission occurs mainly through respiratory droplets (1), the main preventative measures include hand hygiene, social distancing, face masks and quarantine. These measures effectively limit the transmission of SARS-CoV-2 (2) and can also influence the transmission of other common respiratory viruses, including respiratory syncytial virus (RSV), Influenza and seasonal human coronaviruses (HCoVs). RSV is one of the most common viruses to infect infants worldwide and a leading cause of hospital admission (3–5). Infants are also at increased risk of Influenza virus-associated respiratory infections, with the highest hospitalization rates in infants below 6 months of age (6). Less is known about the impact of seasonal HCoVs in infants, but it has also been suggested that these viruses pose a substantial burden of lower respiratory tract infections (7).

Infants are vulnerable to viral infections since their immune system is still developing, with a limited ability to produce an effective antibody response (8). Initially, the infant is partially protected by maternal antibodies transferred through the placenta and later by maternal secretory immunoglobulin A (IgA), secretory immunoglobulin M (IgM), and immunoglobulin G (IgG) through breastfeeding (9, 10). Most antibodies in human milk are produced by plasma cells in the mammary gland. These antibodies are transferred into the milk via receptors on endothelial cells (11). Human milk contains the highest quantity of secretory IgA, but it has been suggested that IgG derived from milk also plays a role in fighting pathogens in the young infant (9). Specifically, anti-RSV IgG in human milk was shown to correlate with a reduced incidence of acute RSV infection in the infant (12). Overall, breastfed infants have fewer respiratory infections and a lower mortality risk compared to infants who received formula (13, 14).

In general, incidence of respiratory viruses is higher during winter. However, during the winter season 2019 to 2020, a decrease of more than 90% in the detection of RSV and Influenza virus has been observed in the Netherlands (15). Subsequently, a massive delayed epidemic occurred during summer 2020, mostly due to RSV infections, resulting in a national capacity crisis at pediatric wards and intensive care units (16). As lactating mothers were less exposed to common viruses during the COVID-19 pandemic (17), we hypothesized that the maternal immune system was not boosted during the pandemic related public health measures, resulting in a lower production of antibodies in maternal serum and human milk. We additionally hypothesized that a COVID-19 infection itself might alter the production of other antibodies due to an overall activation of the immune system (18). Therefore, in this study, we aim to determine the association between the COVID-19 preventative measures and the presence of antibodies against several respiratory viruses in maternal serum and human milk.

## RESULTS

**Reduced IgG levels in human milk during the COVID-19 pandemic.** We investigated respiratory virus-specific antibody levels in milk and serum of in total 114 lactating mothers (Table 1). First, we compared milk IgG and IgA levels at the start of the lockdown (April to May 2020) and 6 months into the lockdown (October-November 2020) (groups 1 and 2, respectively). Four participants from group 1 (and their matches) with a highly suspected SARS-CoV-2 infection were excluded from this comparison as they did not have SARS-CoV-2-specific antibodies in their milk nor serum and were therefore considered not to have been previously infected. We observed that during the lockdown, IgG, but not IgA levels, to RSV, Influenza, HCoV-OC43, HCoV-HKU1, and HCoV-NL63 were significantly reduced in human milk with a factor of 1.7 ($P < 0.001$), 2.2 ($P < 0.01$), 2.6 ($P < 0.05$), 1.4 ($P < 0.01$), and 2.1 ($P < 0.001$), respectively (Fig. 1).

In contrast, maternal serum antibodies increased during the pandemic (Fig. S2, IgG: with a factor of 1.7 ($P < 0.01$) for RSV, 2.0 ($P < 0.05$) for HCoV-229E and 1.5 ($P < 0.05$) for HCoV-NL63. IgA: with a factor of 3.5 ($P < 0.0001$) for RSV, 2.0 ($P < 0.05$) for Influenza, 3.2 ($P < 0.001$) for HCoV-OC43, 3.2 ($P < 0.001$) for HCoV-HKU1 and 2.2 ($P <$

**TABLE 1** Study groups, collection periods, and participant characteristics. Per group, PCR confirmed SARS-CoV-2 infection status, sample collection period, season of collection, and the implemented COVID-19 prevention measures during these collection times in the Netherlands are displayed[a]

| Collection period | Group 1 (n = 38) April 2020 – May 2020 | Group 2 (n = 38) October 2020 – November 2020 | Group 3 (n = 38) October 2020 – November 2020 |
|---|---|---|---|
| Season | Spring | Autumn | Autumn |
| Lockdown | Start | During | During |
| Previous SARS-CoV-2 infection | Yes | Yes | No |
| Age (yrs) Mean ± SD | 31 (± 3.3) | 33 (± 4.2) | 31 (± 3.0) |
| Gestational period (wks) median (IQR) | 39 (38–40) | 40 (39–41) | 40 (39–41) |
| Lactation period at time of milk sample (days) median (IQR) | 194 (76.3–268.8) | 222 (152.3–361.8) | 194 (75.5–268.8) |
| Time positive test and sample (days) median (IQR) | 30 (19.8–42.3) | 37 (26.5–82.8) | Not applicable |

[a]SD = standard deviation, IQR = interquartile range.

0.01) for HCoV-229E). Since this was opposite to our hypothesis, we sought to verify this finding in an additional cohort. We additionally evaluated IgG levels against the same virus antigens in a total of 84 individuals who did not experience COVID-19 during that period. In this group, we did not find differences in IgG levels to the respiratory viruses during the pandemic, except for HCoV-OC43 specific IgG, which was decreased 6 months into the pandemic (Fig. S3).

**Increased IgG and IgA levels in human milk after COVID-19.** Furthermore, our cohort included mothers with and without previous COVID-19 (groups 2 and 3), and thus we also sought to investigate whether the COVID-19 convalescent mothers had different levels of antibodies to these respiratory viruses in their milk. We compared human milk antibody levels in October-November 2020 and found significantly higher IgG levels in human milk of mothers who experienced COVID-19 compared to mothers who did not, for almost all investigated viruses (with a factor of 1.7 ($P < 0.01$) for RSV, 2.5 ($P < 0.001$) for HCoV-OC43, 2.0 ($P < 0.05$) for HCoV-HKU1, 1.9 ($P < 0.001$) for HCoV-229E, and 1.5 ($P < 0.05$) for HCoV-NL63) and significantly higher IgA levels for almost all viruses (with a factor of 4.3 ($P < 0.01$) for RSV, 2.0 ($P < 0.05$) for Influenza, 3.5 ($P < 0.05$) for HCoV-OC43, 2.4 ($P < 0.05$) for HCoV-HKU1 and 2.3 ($P < 0.05$) for HCoV-NL63) (Fig. 2).

## DISCUSSION

In this study, we aimed to determine the influence of the COVID-19 preventative measures on antibody levels against several respiratory viruses, and demonstrated a decrease in IgG levels against RSV, Influenza, HCoV-OC43, HCoV-HKU1, and HCoV-NL63 in human milk but not in serum. Additionally, we observed that women who recently experienced COVID-19 had boosted IgG and IgA levels to other respiratory viruses in human milk.

This study provides insight on the effect of the lockdown and hygiene measures on mother-to-infant transfer of passive immunity via human milk antibodies. Our findings provide a possible explanation for the observed increased incidence of infant hospital admission due to respiratory infections since the restrictions were partially released (16). Knowledge on the passive immunity of breastfed infants during the corona pandemic is important to anticipate for an increase in (severe) respiratory infections and to prepare for capacity problems after the release of the preventative measures.

Interestingly, while we observed a decrease in human milk IgG levels during the pandemic, we observed no difference in IgA levels. The different dynamics in IgA and IgG antibody responses could be explained by their different immune functions. IgA plays a key role in the initial immune response as the first line of defense against the virus and wanes after a relatively short period, while IgG is predominantly important in the secondary immune response and is maintained for months (19). Our first collection time point may have been too late to detect the decrease in IgA levels. Growing

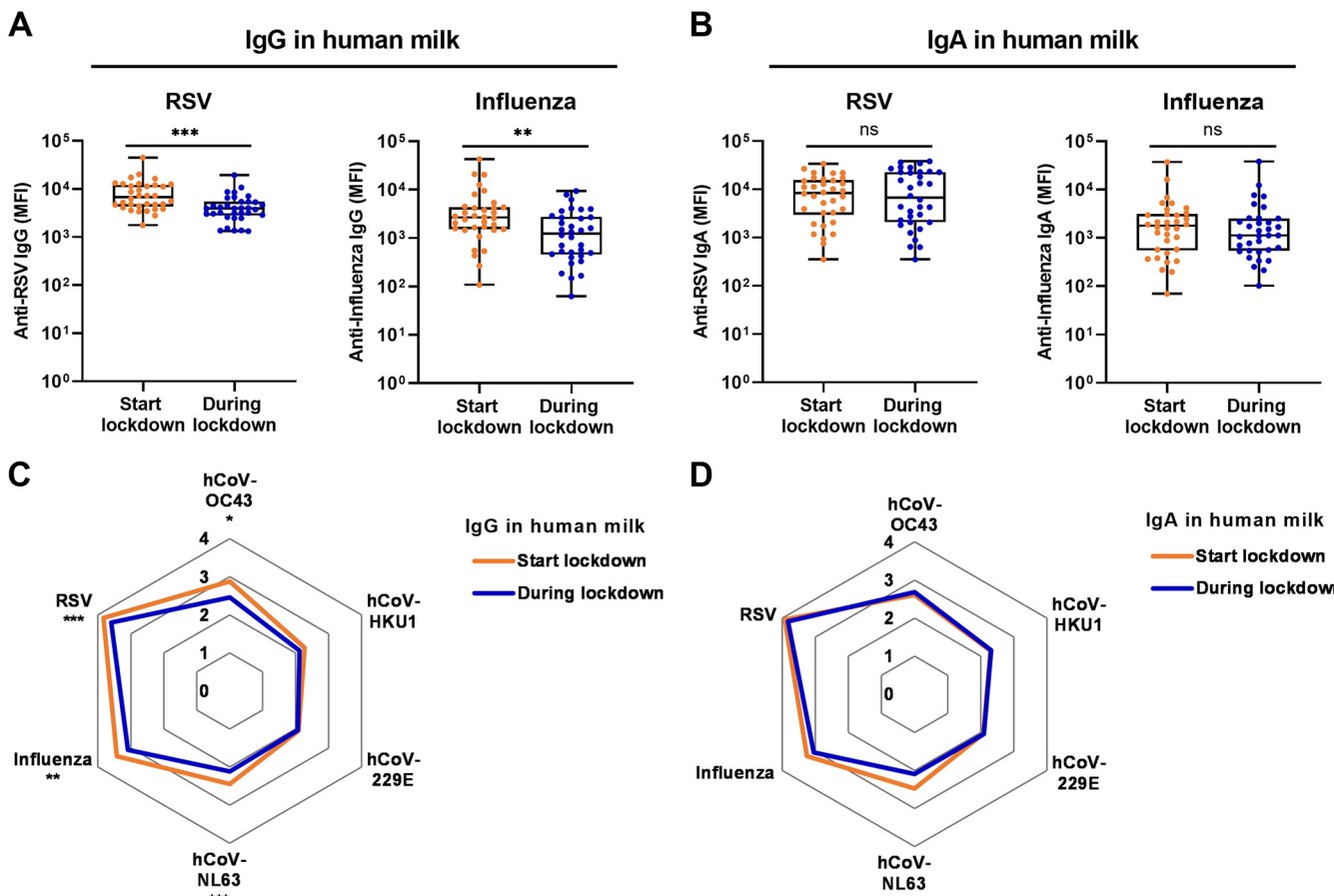

**FIG 1** Levels of respiratory virus antibodies in milk of lactating mothers at the start of the pandemic and during the pandemic. (A) Anti-RSV (left) and anti-Influenza (right) IgG levels in median fluorescence intensity (MFI) in human milk of 34 mothers at the start of the COVID-19 lockdown (April-May 2020, orange dots) were compared to 34 mothers during the COVID-19 lockdown (October-November 2020, blue dots) using a Mann-Whitney U test. Boxplots range the minimum and maximum values. (B) Anti-RSV (left) and anti-Influenza (right) IgA levels in human milk of 34 mothers at the start of the COVID-19 lockdown (April–May 2020, orange dots) were compared to 34 mothers during the COVID-19 lockdown (October-November 2020, blue dots) using a Mann-Whitney U test. Ns = not significant. (C) Spiderweb plot showing IgG levels and (D) IgA levels to all viral antigens as the logarithm of the MFI of each group. The two groups were compared individually per antigen with a Mann-Whitney U test. ***, $P < 0.001$; **, $P < 0.01$; *, $P < 0.05$; ns or no asterisk, not significant.

evidence suggests that human milk IgG also plays an important role in fighting pathogens, and IgG, but not IgA levels, in human milk were associated with protection against RSV infections in infants (9, 12). Although we observed decreased IgG levels in human milk for four out of the six investigated viruses, HCoV-HKU1 and HCoV-229E IgG titers remained stable. Antibody levels against these two viruses were already low at the start of the pandemic, and thus these results are likely explained by the bi-annual seasonality of the seasonal coronaviruses, which would indicate that these two HCoV viruses were not circulating in the winter preceding our study (20).

While decreasing antibody levels were observed in human milk, maternal serum antibody levels increased during the pandemic. We extended this research to a larger cohort with participants without a previous SARS-CoV-2 infection and found that serum antibodies remained stable, except for HCoV-OC43. The contrast observed between serum and milk is an interesting reflection of the difference between antibody production and/or secretion between the two compartments in response to pathogen encounters. Our results indicate that human milk is a dynamic immune organ and suggest that the antibody levels in milk are more affected by lack of immune-boosting than antibody levels in the circulation. What mechanism is responsible for this difference remains an unanswered question. Our findings showed that this process might not be an entirely pathogen-specific effect, as we observed increased antibody levels for almost all investigated viruses in the mothers with previous COVID-19.

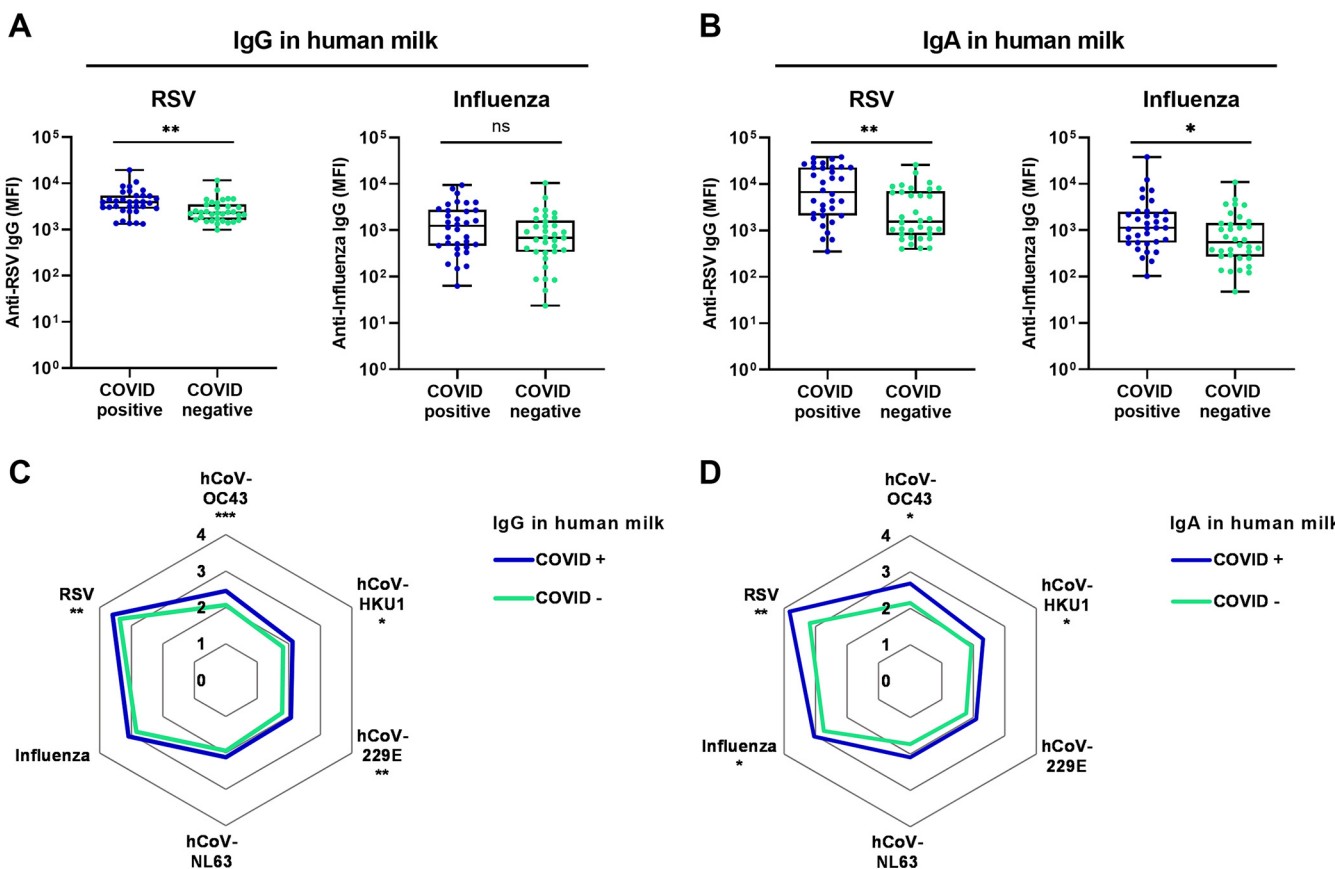

**FIG 2** Levels of respiratory virus antibodies in lactating mothers with recent COVID-19 and without recent COVID-19. (A) Anti-RSV (left) and anti-Influenza (right) IgG levels in median fluorescence intensity (MFI) in human milk of 34 mothers with recent COVID-19 (blue dots) were compared to 34 mothers with no recent COVID-19 (green dots) using a Mann-Whitney U test. Boxplots range the minimum and maximum values. (B) Anti-RSV (left) and anti-Influenza (right) IgA levels in human milk of 34 mothers with recent COVID-19 (blue dots) were compared to 34 mothers with no recent COVID-19 (green dots) using a Mann-Whitney U test. (C) Spiderweb plot showing IgG levels and (D) IgA levels to all viral antigens as the logarithm of the MFI of each group. The two groups were compared individually per antigen with a Mann-Whitney U test. *, $P < 0.05$, **, $P < 0.01$, $P < 0.001$, ns or no asterisk, not significant.

The higher antibody levels to seasonal human coronaviruses observed in milk from individuals who were previously infected with SARS-CoV-2, could be due to antibody cross-reactivity (21, 22). Less expected were the higher antibody levels to RSV and Influenza in the participants with previous COVID-19. For this finding, there are two possible explanations: (i) lactating women that encountered SARS-CoV-2 could be more at risk of encountering viruses in general, due to for example their profession or more social contacts, and thereby have a higher risk of also encountering other respiratory viruses, and (ii) the immune system of the lactating mother is boosted by the SARS-CoV-2 infection, leading to higher antibody levels. It is possible that also the reduced antibody levels observed in human milk during the restrictions should not only be viewed as the absence of immune stimulation with each virus individually but additionally as a general reduction of immune stimulation leading to reduced antibody production or secretion in the milk.

A strength of this study is that participants were matched on duration of lactation, which minimalizes the effects on antibody concentrations between groups (23). Furthermore, human milk collection was standardized, resulting in milk samples of consistent quality and composition between study groups. The Luminex assay allows measurement of antibody levels to several viruses in a single well, reducing introduction of variability. This flexible platform allows investigation of a panel of different viruses and assessment of both IgG and IgA levels in multiple sample types, enabling an in-depth characterization of our limited sample. Possible limitations of this study include the uncertainty concerning the functionality of the measured antibodies. Even

though antibody presence was detected, neutralization capacity and other functions were not studied. Although we analyzed human milk and serum samples from 114 women, the sample size per study group was relatively small, making it harder to generalize our results. Moreover, the retrospective design did not allow us to include a study group with and without a previous SARS-CoV-2 infection in every period. Moreover, it was not possible to correct for the possible additional influence of the different seasons (spring and fall) on antibody levels. However, as several viruses did not circulate during the pandemic, the influence of virus seasonality might be limited.

**Conclusion.** Passive immunity derived from antibodies in human milk is important for protecting young infants against invading viruses. The waning of antibody levels in human milk that we found in this study might partially explain the surge of hospitalizations of infants, mostly due to RSV. We observed a similar profile for other respiratory viruses, including Influenza virus and HCoVs, but this pattern is probably not only limited to these viruses. Knowledge of the association between preventative measures and antibody levels in human milk and subsequently passive immunity in infants might help predict infant hospital admissions and thereby enables anticipation to prevent capacity issues. Additionally, it is important in the consideration for strategies for future lockdowns to best prevent possible consequences for vulnerable infants.

## MATERIALS AND METHODS

**Study design.** During the COVID-19 pandemic, over 2500 human milk samples were collected in different studies conducted by the Amsterdam University Medical Centre (24, 25). The 114 samples for the current study were selected out of these 2500 samples. At the start of the pandemic, samples were collected from 38 lactating mothers with a previous PCR (PCR) confirmed or highly suspected SARS-CoV-2 infection (25). From the larger cohort, two groups were matched to this first group in order to answer the following research questions: (i) what is the influence of the preventative measures on human milk antibody levels against common respiratory viruses? (ii) what is the influence of a previous SARS-CoV-2 infection on human milk antibody levels? The samples in group 1 were collected between April 2020 and the end of May 2020 (Table 1). This period coincided with the start of the first lockdown and the implementation of preventative measures to limit the spread of SARS-CoV-2 (Fig. S1). The participants in group 2 also had a previous PCR-confirmed SARS-CoV-2 infection and samples were collected between October 2020 until the end of November 2020, after the summer during the COVID-19 restrictions. To be able to solely determine the influence of a SARS-CoV-2 infection on human milk antibody levels against common respiratory viruses, another group without a previous SARS-CoV-2 infection was selected in the same period; group 3. Samples in each group were matched (1:1:1) on the following criteria: duration of lactation at the time of sample collection and time in days between the date of a PCR (PCR) confirmed SARS-CoV-2 infection and the date of sample collection. Written informed consent to use their characteristics and samples for future research was obtained from all participants.

Additionally, to compare with the results from maternal serum in our 114 participants, 133 serum samples from 84 health care workers were provided by the prospective serologic surveillance cohort (S3-study) conducted in two tertiary care medical centers in the Netherlands (26). In short, participants were included in the cohort from the beginning of the COVID-19 pandemic in the Netherlands in March 2020. Participants donated blood every 4 weeks from March until June 2020 and again in October 2020, which was tested for presence of SARS-CoV-2 antibodies, and completed a questionnaire about COVID-19 related symptoms and SARS-CoV-2 PCR test results. Only SARS-CoV-2 seronegative participants were included in our analysis, and only the samples collected in April-May 2020 ($n = 82$) and October-November 2020 ($n = 51$), corresponding with the periods in which the mothers also donated serum and milk. The local ethical committee approved the study in both hospitals, and all participants gave written informed consent before participating, including consent to determine antibody levels against other viruses (NL73478.029.20).

**Sample collection.** Human milk samples were collected in the morning before the first feeding moment. Participants were requested to empty one breast, mix the fore- and hind milk and then donate in a sterile tube. Samples were collected either at the participant's home or the research location. Samples were subsequently stored at $-80°C$ pending analysis. On the same day as human milk collection, serum was collected and stored at $-80°C$ pending analysis.

**Laboratory analysis: antigens.** To assess antibody levels against the different respiratory viruses, we designed and produced their main virus antigens, all class I fusion proteins. Prefusion stabilized trimeric spike proteins of HCoV-OC43, HCoV-HKU1, HCoV-229E, and HCoV-NL63, prefusion stabilized trimeric RSV-fusion glycoprotein (Strain A2, design: [27]), and trimeric Influenza hemagglutinin (H1N1$_{pdm2009}$ A/Netherlands/602/2009, design: [28]) were produced in HEK293F cells and purified by NiNTA chromatography followed by size exclusion chromatography as described previously (29).

**Laboratory analysis: assays.** Antibody levels in serum and milk were assessed using a custom Luminex assay similar to previously described (29). In short, viral antigens were covalently coupled to Luminex Magplex beads with a two-step carbodiimide reaction at a ratio of 37 $\mu$g protein to 12.5 million beads for RSV-F glycoprotein and equimolar amounts for other proteins. Following the outcome of

optimization experiments, serum was diluted 1:10,000, and milk was diluted 1:100. Beads and diluted samples were incubated overnight, followed by detection with goat-anti-human IgG-PE or goat-anti-human IgA-PE (Southern Biotech). Read-out was performed on a Magpix (Luminex). The resulting median fluorescence intensity (MFI) values are the median of at least 50 beads per well and were corrected by subtraction of MFI values from the buffer and beads-only wells.

**Data analysis.** *P*-values below 0.05 were considered significant. We compared median antibody titers between group 1 and 2 for each antigen in serum and milk separately and we compared median antibody titers between group 2 and 3 for each antigen in milk. We performed Mann-Whitney U-tests for all unpaired comparisons involving two groups in GraphPad Prism 8.3.0. Spiderweb plots were generated in Excel 2016. To compare groups from the S3-study, Linear Mixed Models were used for paired comparisons using IBM SPSS statistics 26 for Windows and the results were visualized using GraphPad Prism 8.3.0.

## SUPPLEMENTAL MATERIAL

Supplemental material is available online only.

**SUPPLEMENTAL FILE 1**, PDF file, 0.7 MB.

## ACKNOWLEDGMENTS

We thank Tim Beaumont for manuscript editing and all participating mothers and health care workers for their contribution to this study.

This work was supported by Stichting Steun Emma Kinderziekenhuis. M.J.vG. acknowledges the Amsterdam Infection and Immunity Institute for funding this work through the COVID-19 grant [24175]. Funding for the S3 study was provided by ZonMw; and the Amsterdam UMC Corona Research Fund.

J.B.vG. is the founder and director of the Dutch National Human Milk Bank and a member of the National Health Council. J.B.vG. has been a member of the National Breastfeeding Council from March 2010 to March 2020.

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
