## [Reviewer comments · Microbiology Spectrum]

Microbiology Spectrum

Decreased passive immunity to respiratory viruses through human milk during the COVID-19 pandemic

Marloes Grobden, Hannah Juncker, Karlijn van der Straten, Ayesha Lavell, Michiel Schinkel, David Buis, Maarten Wilbrink, Khadija Tejjani, Mathieu Claireaux, Aafke Aartse, Christianne De Groot, Dasja Pajkrt, Marije Bomers, Jonne Sikkens, Marit van Gils, Hans van Goudoever, and Britt Van Keulen

Corresponding Author(s): Hans van Goudoever, Emma Children's Hospital

Review Timeline:

Submission Date:	February 1, 2022
Editorial Decision:	April 22, 2022
Revision Received:	May 31, 2022
Accepted:	June 11, 2022

Editor: Yongjun Sui

Reviewer(s): Disclosure of reviewer identity is with reference to reviewer comments included in decision letter(s). The following individuals involved in review of your submission have agreed to reveal their identity: James N Moy (Reviewer #1); Christopher L Chase (Reviewer #3)

Transaction Report:

DOI: <https://doi.org/10.1128/spectrum.00405-22>

April 22, 2022

Prof. Hans B van Goudoever
Emma Children's Hospital
Pediatrics
De Boelelaan 1117
Amsterdam
Netherlands

Re: Spectrum00405-22 (Decreased passive immunity to respiratory viruses through human milk during the COVID-19 pandemic)

Dear Prof. Hans B van Goudoever:

Link Not Available

Sincerely,

Yongjun Sui

Journals Department
Reviewer comments:

Reviewer #1 (Comments for the Author):

The authors of this manuscript hypothesized that, as a result of public health measure aimed at containing the spread of COVID-19 during the SARS-CoV-2 pandemic, antibody levels in human milk are lower during the pandemic and can lead to decrease protection of infants against viral respiratory infections when COVID-19 mitigation efforts are relaxed. The investigators assayed milk antibody levels in 3 groups of women: in group 1 (SARS-CoV-2 positive), samples were collected in April and May of 2020 (at the start of lockdown measures); in group 2 (SARS-CoV-2 positive), samples were collected in October and November of 2020 (6 months into the lockdown); group 3 were SARS-CoV-2 negative, samples collected in October and November 2020.

This study showed that IgG (but not IgA) levels to RSV, influenza and 3 common cold coronaviruses were significantly lower in group 2 than in group 1. On the other hand, group 2 had significant increases in serum IgG and IgA to the respiratory viruses compared to Group 1. When compared to lactating women who were SARS-CoV-2 negative (group 3), SARS-CoV-2 positive women (Group 2) had higher milk IgG and IgA levels for most of the respiratory viruses tested.

Comment

On page 7, lines 132 and 133, the authors stated that 4 participants in group 1 and their matched controls were excluded from analysis because the 4 participants did not have SARS-CoV-2 specific antibodies in their milk. Did these 4 participants have SARS-CoV-2 antibodies in their blood? Are the authors implying that these 4 participants were not truly infected with SARS-CoV-2? The authors discussed (page 9, lines 184 through 191) possible reasons why antibody levels are higher in blood than in milk. Because the participants in group 1 had positive SARS-CoV-2 PCR tests, all of the participants should be included in the analysis. I would like to see the results of the statistical analysis performed with all participants in groups 1 and 2.

Reviewer #2 (Comments for the Author):

Grobben and colleagues describe the findings of a study assessing antibody titres in breast milk and serum from mothers with and without documented SARS-CoV-2 infection at two time points in 2020.

The manuscript is generally well written, however limitations in the study design mean that definitive conclusions are difficult to draw.

Major suggestion

1) The time points chose for the pre and post assessment cover a relatively short period, which co-incides with the summer period in the Netherlands, where RSV and influenza have low circulation. As such, it is difficult to infer what additional impact the COVID-19 restrictions have had over and above any usual inter-seasonal waning of antibodies. Further sampling into mid-2021 (prior to the delayed surge in RSV) would allow for more robust assessment of the impact of the delayed RSV and influenza season on antibody titres.

Minor suggestions:

Introduction

1) line 39 -consider a reference for poor infant antibody response
<https://www.ncbi.nlm.nih.gov/pmc/articles/PMC4707740/>

2) line 47 - careful on quantification of impact on mortality for breastfeeding - 2.2 fold lower risk - in the European context this is unlikely to be true (but in the developing world certainly - where most of the studies reviewed in the reference were conducted) - perhaps:
"and reduced infant mortality, a difference which is most marked in developing settings"

3) line 56 - use "public health" rather than "quarantine"

4) line 58 - clarify what is meant by "hyper activation of immune responses" and provide a reference? Is there documentation of activation of memory B-cells to respiratory viruses following COVID-19 infection?

Methods

1) greater clarity around the groups would be useful - I would avoid using the term "control" groups as this is confusing - perhaps just cohort 1, cohort 2 and cohort 3 - with a description of timing of collection & SARS-CoV-2 infections status for each group.

2) the inclusion of the health care worker cohort here is confusing - i believe as this was not part of the original study design - perhaps give the context of inclusion of analysis of these samples in the methods (i.e. to compare with changes in maternal serum IgG between the two time points)

3) consent - was this study set up as a biobank? did participants give consent for samples to be used in this specific study (or a previous study and subsequent studies including this one?)

4) line 121 - state here which groups were compared in the analysis and what was compared (antibody titer levels presumably)

Results

1) line 130 - I would open here with the total number of samples (114 with 38 from each cohort?) and refer to table 1

2) state if comparing mean titres? either here or in the methods

Reviewer #3 (Comments for the Author):

This is a well written paper that is easy to read and understand. The design and conclusions are solid.

Staff Comments:

Preparing Revision Guidelines

Please return the manuscript within 60 days; if you cannot complete the modification within this time period, please contact me. If you do not wish to modify the manuscript and prefer to submit it to another journal, please notify me of your decision immediately so that the manuscript may be formally withdrawn from consideration by Microbiology Spectrum.

Decreased passive immunity to respiratory viruses through human milk during the COVID-19 pandemic\
Microbiology Spectrum

The authors of this manuscript hypothesized that, as a result of public health measure aimed at containing the spread of COVID-19 during the SARS-CoV-2 pandemic, antibody levels in human milk are lower during the pandemic and can lead to decrease protection of infants against viral respiratory infections when COVID-19 mitigation efforts are relaxed. The investigators assayed milk antibody levels in 3 groups of women: in group 1 (SARS-CoV-2 positive), samples were collected in April and May of 2020 (at the start of lockdown measures); in group 2 (SARS-CoV-2 positive), samples were collected in October and November of 2020 (6 months into the lockdown); group 3 were SARS-CoV-2 negative, samples collected in October and November 2020. This study showed that IgG (but not IgA) levels to RSV, influenza and 3 common cold coronaviruses were significantly lower in group 2 than in group 1. On the other hand, group 2 had significant increases in serum IgG and IgA to the respiratory viruses compared to Group 1. When compared to lactating women who were SARS-CoV-2 negative (group 3), SARS-CoV-2 positive women (Group 2) had higher milk IgG and IgA levels for most of the respiratory viruses tested.

Comment

On page 7, lines 132 and 133, the authors stated that 4 participants in group 1 and their matched controls were excluded from analysis because the 4 participants did not have SARS-CoV-2 specific antibodies in their milk. Did these 4 participants have SARS-CoV-2 antibodies in their blood? Are the authors implying that these 4 participants were not truly infected with SARS-CoV-2? The authors discussed (page 9, lines 184 through 191) possible reasons why antibody levels are higher in blood than in milk. Because the participants in group 1 had positive SARS-CoV-2 PCR tests, all of the participants should be included in the analysis. I would like to see the results of the statistical analysis performed with all participants in groups 1 and 2.

We appreciate the time and effort the reviewers spent to review our manuscript. We are thankful for the constructive feedback, which will improve the quality of the manuscript. We tried to address each of the reviewers comments, our replies are in blue.

Reviewer comments:

Reviewer #1 (Comments for the Author):

The authors of this manuscript hypothesized that, as a result of public health measure aimed at containing the spread of COVID-19 during the SARS-CoV-2 pandemic, antibody levels in human milk are lower during the pandemic and can lead to decrease protection of infants against viral respiratory infections when COVID-19 mitigation efforts are relaxed. The investigators assayed milk antibody levels in 3 groups of women: in group 1 (SARS-CoV-2 positive), samples were collected in April and May of 2020 (at the start of lockdown measures); in group 2 (SARS-CoV-2 positive), samples were collected in October and November of 2020 (6 months into the lockdown); group 3 were SARS-CoV-2 negative, samples collected in October and November 2020. This study showed that IgG (but not IgA) levels to RSV, influenza and 3 common cold coronaviruses were significantly lower in group 2 than in group 1. On the other hand, group 2 had significant increases in serum IgG and IgA to the respiratory viruses compared to Group 1. When compared to lactating women who were SARS-CoV-2 negative (group 3), SARS-CoV-2 positive women (Group 2) had higher milk IgG and IgA levels for most of the respiratory viruses tested. We thank the reviewer for the accurate summary of our work. Please find below the point-by-point reply and adjustments we have made in response to the comments and suggestions.

Comment

On page 7, lines 132 and 133, the authors stated that 4 participants in group 1 and their matched controls were excluded from analysis because the 4 participants did not have SARS-CoV-2 specific antibodies in their milk. Did these 4 participants have SARS-CoV-2 antibodies in their blood? Are the authors implying that these 4 participants were not truly infected with SARS-CoV-2?

In our previous study, we included 38 participants with a previous Polymerase Chain Reaction (PCR) confirmed or highly suspected SARS-CoV-2 infection. They were classified as highly suspected if someone in their household had a positive PCR test and they themselves had COVID-19 symptoms. The 4 participants that did not have antibodies in their milk, were amongst the highly suspected participants and indeed also did not show antibodies in their serum. As they did not have a proven PCR positive result and no antibodies, we could not confirm infection with SARS-CoV-2 and therefore excluded these participants. We clarified this in the method and results section (lines 74-75 and 142-144).

The authors discussed (page 9, lines 184 through 191) possible reasons why antibody levels are higher in blood than in milk. Because the participants in group 1 had positive SARS-CoV-2 PCR tests, all of the participants should be included in the analysis. I would like to see the results of the statistical analysis performed with all participants in groups 1 and 2.

As stated in the comment above, not all participants in group 1 did actually have a positive SARS-CoV-2 PCR test. This is now explained better in the methods section (lines 74-75).

Reviewer #2 (Comments for the Author):

Grobben and colleagues describe the findings of a study assessing antibody titres in breast milk and serum from mothers with and without documented SARS-CoV-2 infection at two time points in 2020.

The manuscript is generally well written, however limitations in the study design mean that definitive conclusions are difficult to draw.

We would like to thank the reviewer for reviewing our manuscript and the constructive comments.

Major suggestion

1) The time points chose for the pre and post assessment cover a relatively short period, which co-incides with the summer period in the Netherlands, where RSV and influenza have low circulation. As such, it is difficult to infer what additional impact the COVID-19 restrictions have had over and above any usual inter-seasonal waning of antibodies. Further sampling into mid-2021 (prior to the delayed surge in RSV) would allow for more robust assessment of the impact of the delayed RSV and influenza season on antibody titres.

The duration between our time points was 6 months, during which COVID-19 lockdown measures were in place. We chose these time points as some of the lockdown measures were released after October 2020 and we expect that measurements at these chosen time points enabled us to answer our research question. However, the reviewer raises a good point of discussion that the influence of seasonality could not be taken into account in this way. We addressed this point in our limitation section (lines 227-231). It would have been interesting to analyze milk samples from October 2019, but unfortunately we do not have these samples available. Likewise, we do not have samples available from May 2021 and also we expect that antibody titers in samples from this time period could additionally be influenced by both release of some lockdown measures and by the start of the vaccination program in the Netherlands. All of this is discussed in the limitations section in the discussion.

Minor suggestions:

Introduction

1) line 39 -consider a reference for poor infant antibody response

<https://www.ncbi.nlm.nih.gov/pmc/articles/PMC4707740/>

The reference has been added to line 46.

2) line 47 - careful on quantification of impact on mortality for breastfeeding - 2.2 fold lower risk - in the European context this is unlikely to be true (but in the developing world certainly - where most of the studies reviewed in the reference were conducted) - perhaps:

"and reduced infant mortality, a difference which is most marked in developing settings"

We have adjusted it into "Overall, breastfed infants have fewer respiratory infections and a lower mortality risk compared to infants who received formula."

3) line 56 - use "public health" rather than "quarantine"

This is a valid suggestion and we have replaced “quarantine” with “public health”.

4) line 58 - clarify what is meant by "hyper activation of immune responses" and provide a reference? Is there documentation of activation of memory B-cells to respiratory viruses following COVID-19 infection?

Indeed ‘hyper activation of immune responses’ was not very clear and we have changed this to ‘overall activation of the immune system’ and provided a reference on line 64.

Methods

1) greater clarity around the groups would be useful - I would avoid using the term "control" groups as this is confusing - perhaps just cohort 1, cohort 2 and cohort 3 - with a description of timing of collection & SARS-CoV-2 infections status for each group.

We agree that the annotation “control” is confusing. However, we think it should be clear that groups 2 and 3 were matched to group 1, as that is what limits our sample size. We have adjusted this section to make it more clear in the text (lines 72-75) and Table 1.

2) the inclusion of the health care worker cohort here is confusing - i believe as this was not part of the original study design - perhaps give the context of inclusion of analysis of these samples in the methods (i.e. to compare with changes in maternal serum IgG between the two time points)

Indeed the explanation of the inclusion of the health care workers was not completely clear. We have added the reasoning to add this cohort in our study to this section on line 90.

3) consent - was this study set up as a biobank? did participants give consent for samples to be used in this specific study (or a previous study and subsequent studies including this one?)

Our participants were recruited from previous studies. When they gave consent for storing their samples in a biobank and the use for future research, they were eligible for this specific sub study. We have added this in lines 88-89 and 99-101.

4) line 121 - state here which groups were compared in the analysis and what was compared (antibody titer levels presumably)

We have added which groups we compared for which sample types and antigens (lines 130-132).

Results

1) line 130 - I would open here with the total number of samples (114 with 38 from each cohort?) and refer to table 1

We have added a sentence referring to table 1 that mentions the total number of samples analyzed in the study at the beginning of the paragraph (lines 139-140).

2) state if comparing mean titres? either here or in the methods

We compare median titers, this has now been noted in the methods (lines 130-132).

Reviewer #3 (Comments for the Author):

This is a well written paper that is easy to read and understand. The design and conclusions are solid.

We thank the reviewer for the compliments.

June 11, 2022

Prof. Hans B van Goudoever
Emma Children's Hospital
Pediatrics
De Boelelaan 1117
Amsterdam
Netherlands

Re: Spectrum00405-22R1 (Decreased passive immunity to respiratory viruses through human milk during the COVID-19 pandemic)

Dear Prof. Hans B van Goudoever:

Your manuscript has been accepted, and I am forwarding it to the ASM Journals Department for publication. You will be notified when your proofs are ready to be viewed.

Sincerely,

Yongjun Sui
Editor, Microbiology Spectrum
